# A Novel Variable-Proportion Desaturation PI Control for Speed Regulation in Sensorless PMSM Drive System

**Zihan Wei** , **Mi Zhao \*** , **Ximu Liu** and **Min Lu**

College of Mechanical and Electrical Engineering, Shihezi University, Shihezi 832003, China; weizihan615@163.com (Z.W.); lzy3301357367@163.com (X.L.); lm_shz@163.com (M.L.)
* Correspondence: zhaomi@shzu.edu.cn

**Featured Application: Elimination of integral saturation and optimization of dynamic performance of sensorless drive system of permanent magnet synchronous motor based on SMO-PLL observation.**

**Abstract:** The problem of integral saturation greatly restricts the engineering application of PID controller in AC speed regulator systems, which also affects the control performance of sensorless drive system with the phase-locked loop (PLL) structure. To address this, this paper proposes a novel variable-proportion desaturation PI (VPDPI) control method for permanent magnet synchronous motor (PMSM) sensorless drive system. Firstly, an overall scheme of sensorless control system with sliding mode observer (SMO)-PLL observer is presented, which includes the equivalent mathematical model and its parameter identification model of PMSM. Secondly, the control principle of the VPDPI algorithm is innovatively designed. Note that the novel regulator scheme consists of two interconnected components in terms of the concepts of threshold segmentation, i.e., the variable-proportion link and desaturation PI part. Meanwhile, the stability analysis of the regulator is further discussed by the root locus and the Bode diagram analysis. Finally, the numerical simulations of speed regulation are carried out under the various operation conditions, and the effectiveness of the proposed method is further verified by experimental platform. Several groups of comparative experiments reveal that the proposed method has a more higher control performance for the sensorless drive system, especially in the ability of overshoot suppression.

**Keywords:** permanent magnet synchronous motor (PMSM); sensorless; variable-proportion desaturation PI (VPDPI); integral saturation; observation performance



## 1. Introduction

Permanent magnet synchronous motor (PMSM) has been widely used in electric vehicle, ship electric propulsion system and other driving fields because of its advantages of simple structure, high power density and high operating efficiency [1,2]. With the development of electronic technology, PMSM drive structures are also constantly updated to adapt to the needs of different operating conditions [3,4]. In the detection of traditional drive system, the installation position of mechanical sensors such as photoelectric encoder and rotary transformer is not accurate. In addition, the aging of sensors restricts the drive system to achieve satisfactory control performance, which reduces the system reliability and economics of production [5,6]. In order to solve the influence of traditional mechanical sensors on the drive system, the sensorless idea is developed and applied in PMSM drive system. PMSM sensorless drive system can construct the rotor position and speed information required by detecting the voltage and current signals of the motor port [7,8].

At present, the sensorless control technology of permanent magnet synchronous motor can be roughly divided into two types based on fundamental wave observation and salient pole effect [9,10]. The observation method based on salient pole effect has better

observation performance for low speed operation condition [11,12]. However, it is a pity that this method has some limitations for hidden pole machine [13,14]. Therefore, the method based on fundamental wave observation is still used to obtain PMSM rotor information in more control occasions. The fundamental wave observation method is divided into four types: arc-tangent method, phase-locked loop (PLL) method, model reference adaptive method and intelligent algorithm according to the different methods of obtaining rotor information [15,16]. Among them, the model reference adaptive method has the characteristics of simple structure. However, the accuracy of the observation is closely related to the design rules of the adaptive law, which is difficult to be applied in practical systems [17,18]. In addition, it is difficult to realize in the actual system. Meanwhile, intelligent algorithm observation rotor information accuracy is closely related to the number of training samples, the control structure is very complex [19,20]. On the contrary, the arc-tangent method can use the sliding mode variable structure method to obtain the inverse electromotive force information of the motor rotor, and directly construct the arc-tangent equation to obtain the required position information [21,22]. However, this method has a low signal noise ratio (SNR) and is greatly affected by back EMF harmonics [23,24].

Fortunately, the closed-loop feedback of rotor back EMF information based on phase-locked loop method is constructed by introducing PI controller [25,26]. In the presence of harmonics, closed-loop self-adjustment process can be adopted to alleviate the disadvantages of arc-tangent method [27,28]. It can be seen that the phase-locked loop method is an observation method with relatively good comprehensive performance. Meanwhile, PI regulator introduced by phase-locked loop method also aggravates the problem of integral saturation and integral drift of the whole system. Therefore, in order to achieve better performance of the sensorless system, it is necessary to perform integral saturation suppression on the driving system. In terms of integral saturation suppression, most engineering uses anti-saturation methods, such as variable speed integration and integral separation, to suppress the integral saturation state of the drive system [29–31]. In [32], the influence of saturation phenomenon has been suppressed to a certain extent, nonetheless it does not eliminate the accumulated integral error. Hence, the anti-saturation method is easy to fall into integral saturation again. In order to completely eliminate the integral saturation of the system, the desaturated PI control structure is developed based on the idea of anti-saturation [33–35]. The method can completely remove the integral from the saturation state through the reverse accumulation of errors. However, it should be noted that if the reverse error accumulation is too large, it is easy to weaken the role of the integral link, so that the drive system has steady-state error and increases the response time of the system.

In order to make PMSM sensorless system have better observation and drive performance, the original contributions of this paper are shown as follows:

(1) Compared with the traditional model parameter equivalence, this paper uses MATLAB/System Identification toolbox to realize the parameter identification of high-order system, and the identification accuracy can reach 91.22%. Meanwhile, root locus and Bode diagram are used to complete the stability analysis of the closed-loop system and the selection of VPDPI regulator parameters.

(2) In terms of the ideal response curve as the design objective, this paper analyzes the causes of system saturation phenomenon. Moreover, the influence of saturation phenomenon on dynamic and steady performance of sensorless system is considered in this paper, especially the ability to suppress overshoot, and the VPDPI regulator is innovatively proposed. Meanwhile, the control rules of the proposed regulator are further given.

(3) Simulations and experiments under various working conditions are presented in detail, and the effectiveness and reasonability of the proposed method are both amply verified.

In Section 2, the overall structure and mathematical model of PMSM sensorless drive system are presented step by step. In Section 3, the requirements of ideal response curve are

reviewed, and the design rules and threshold parameter selection rules of VPDPI regulator are given in detail. Section 4 provides the comprehensive test results of the proposed method on MATLAB/Simulink experimental platform. Meanwhile, the validity of the proposed method on hardware platform is further verified in Section 5. Finally, the brief conclusions are drawn in Section 6.

## 2. The Overall Structure of PMSM Sensorless Drive System

### 2.1. Equivalent of PMSM Mathematical Model

It is well known that PMSM sensorless system is composed mainly of the motor ontology, the inverse model, the current controller and the speed controller. Accurate equivalence of each component model is the basis of stability control system.

Generally speaking, the voltage equation of the PMSM in the $d - q$ rotary transformer can be expressed as (1):

$$\begin{cases} u_d = Ri_d + L_d \frac{di_d}{dt} - \omega_e L_q i_q \\ u_q = Ri_q + L_q \frac{di_q}{dt} + \omega_e (L_d i_d + \psi_f) \end{cases} \tag{1}$$

Among them, $u_d$ and $u_q$ are the $d - q$ axis component of the stator voltage. $i_d$ and $i_q$ are the $d - q$ axis component of the stator current; $R$ is the stator resistance. $\omega_e$ is electric angular velocity. $L_d$ and $L_q$ are the inductance component of $d - q$ axis. $\psi_f$ stands for permanent magnet flux.

When the $i_d = 0$ control strategy is used, the voltage equation can be reduced as shown in (2):

$$\begin{cases} u_d = -\omega_e L_q i_q \\ u_q = Ri_q + L_q \frac{di_q}{dt} + \omega_e \psi_f \end{cases} \tag{2}$$

When there is no external load disturbance, the torque equation and the motion equation of the motor will be written as [36]:

$$\begin{cases} J \frac{d\omega_m}{dt} = T_e + T_L - B\omega_m \\ T_e = \frac{3}{2} P_n i_q \psi_f \end{cases} \tag{3}$$

In (3), $\omega_m$ is mechanical angular velocity. $B$ is the damping coefficient. $J$ is usually thought of as the moment of inertia. $T_e$ and $T_L$ are electromagnetic torque and load torque, respectively. $P_n$ is polar logarithm of PMSM.

After the laplace-transform, the transfer function of the PMSM electrical part $G_{PMSMe}$ and mechanical part $G_{PMSMm}$ can be equivalent to:

$$\begin{cases} G_{PMSMm}(s) = \frac{1}{L_q s + R} \\ G_{PMSMm}(s) = \frac{1}{Js + B} \end{cases} \tag{4}$$

It is obviously that PMSM are a second order system composed of two inertial processes.

Meanwhile, the phase voltage of the inverter module output is not in response to the input vector pulse synchronization, and the inverter does not do the voltage amplitude scale transformation. Hence, the inverter gain $K_{in} = 1$, and it is the initial equivalent of the driving part:

$$G_{SVPWM}(s) = \frac{K_{in}}{T_s s + 1} \tag{5}$$

where $T_s$ is defined the switching frequency.

Unfortunately, the inverter is generally used to switch dead areas and delay in action. Therefore, the small inertia link is compensated for (5). The transfer function for an inertia element can be written:

$$G_{ine}(s) = \frac{1}{t_d s + 1} \tag{6}$$

In (6), $t_d$ is the inertia time constant.

If the current controller is equivalent to a differential link and an integral link, the transfer function block diagram of PMSM sensorless system can be expressed in Figure 1.

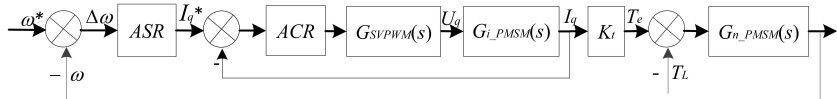

**Figure 1.** The transfer function block diagram of PMSM sensorless system.

It is worth noting that the speed regulator control object $G_{obj}(s)$ is made up of five poles and a zero. Among them, *ASC* and *ACR* are the automatic speed regulator and the automatic current regulator, respectively.

### 2.2. Parameter Identification of the Drive System

In order to better fit the actual mathematical model of the sensorless system, the sensorless system is equivalent to a higher-order system in this paper. However, due to the appropriate condition neglect and approximation in the equivalent process, there is a certain gap between the transfer function obtained directly according to the parameters and the real value. Therefore, MATLAB system identification toolbox can be utilized to identify the known parameters of the zero-pole system. Figure 2 is used as the equivalent control circuit for identifying the unknown parameters.

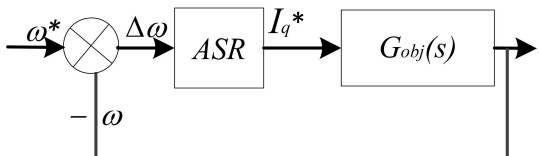

**Figure 2.** The equivalent control circuit.

Where $G_{obj}(s) = \frac{k(s-z)}{\prod(s-p_i)} (i = 1 \ldots 5)$.

General steps to realize unknown parameter Identification of equivalent control object using MATLAB/System Identification toolbox:

Step 1: obtaining identification input signal $I_{qref}$ and output signal $\omega_m$, as shown in Figure 3;

Step 2: calling toolbox and import data;

Step 3: configuring the zeros and poles of the identification system (In terms of Section 2.1, the number of poles = 5, and the number of zeros = 1);

Step 4: running and analyzing the feasibility of identification results.

On the basis of the same input data, the Zero-Pole-Gain (ZPK) model of the post-identification controlled system is shown as (7), and the response is shown in Figure 4.

$$\begin{cases} z = -10.4774 \\ p = [-13405, -16.77, -92.5 \pm 368.8i, -0.0015] \\ k = 8.0228 \times 10^{11} \end{cases} \tag{7}$$

From the response curve estimated by using "sysinit" as template, it can be seen that the similarity between the response and the actual data after identification parameters is 91.22%.

Generally, it can be utilized as the equivalent transfer function of the control object in the parameter debugging process of the controller.

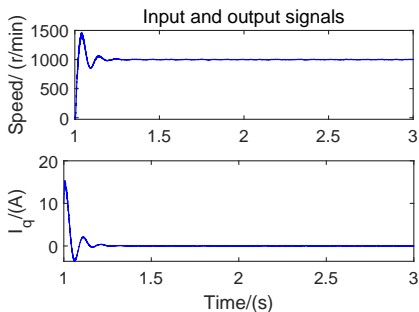

**Figure 3.** The output and input data of the system to be identified.

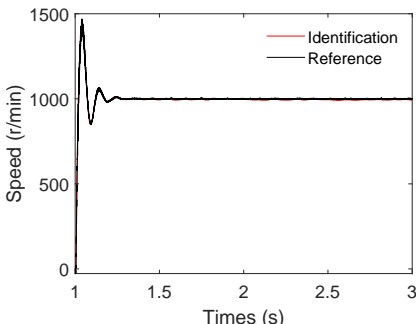

**Figure 4.** Identification result response curve.

### 2.3. The Observer Design of the Rotor Speed and Position Angle

Since the sensorless observation of arctangent method is greatly affected by the back electromotive force, the observation accuracy of position and speed is not ideal. Therefore, this paper adopts phase-locked loop to ensure the accuracy of estimation of rotor speed and rotor position Angle [37].

The construction of reliable sliding surface is the basis of effective sensorless realization. Based on PMSM voltage equation in (1), the observed current equation is constructed as follows:

$$\begin{cases} \frac{d\tilde{i_d}}{dt} = \frac{1}{L_d}(-R\tilde{i_d} + u_d + L_q\omega_e\tilde{i_q} - V_d) \\ \frac{d\tilde{i_q}}{dt} = \frac{1}{L_q}(-R\tilde{i_q} + u_q + L_d\omega_e\tilde{i_d} - V_q) \end{cases} \tag{8}$$

where $V_d$ and $V_q$ are defined the observed electromotive force .

If the observed electromotive force is constructed by (9), it can be used as the input of the PLL [37].

$$\begin{cases} V_d = ksgn(\tilde{i_d} - i_d) \\ V_q = ksgn(\tilde{i_q} - i_q) \end{cases} \tag{9}$$

Hence, the SMO-PLL structure is shown in Figure 5.

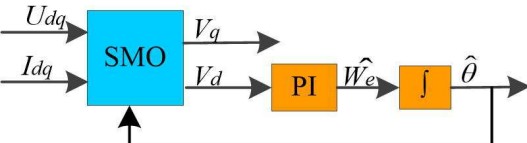

**Figure 5.** The structural diagram of SMO-PLL technology.

It can not be ignored that the phase-locked loop uses PI controller to achieve accurate position Angle tracking, but also aggravates the system integral saturation phenomenon, which will enhance the speed control performance of system overshoot and stability.

### 3. Proposed VPDPI Speed Regulator

*3.1. The Design of the VPDPI Regulator*

There is no doubt that the proportion link can quickly amplify the speed error signal, and it can quickly adjust the speed to the reference value. Unfortunately, the single proportion link will make the system have a certain steady-state error, which can not meet the performance requirements of the most control systems. Fortunately, the integration link is extremely effective in eliminating static errors. However, the control system has a new integral saturation problem since the accumulation of error signals, which leads to the increase of overshoot and even out of control. In order to ensure the control performance of PI controller, it is highly necessary to eliminate the integral saturation phenomenon in the integral link. At present, the anti-saturation ideas is widely used to suppress integral saturation phenomenon, which weakens the negative impact of saturation problem but has not been fundamentally solved. Different from the traditional controller design ideas, a new regulator designed in this paper is based on thoroughly eliminating integral saturation phenomenon and the actual performance requirements of fast response. The ideal response curve with fast response and no overshoot is shown in Figure 6. Therefore, a novel VPDPI regulator is further designed.

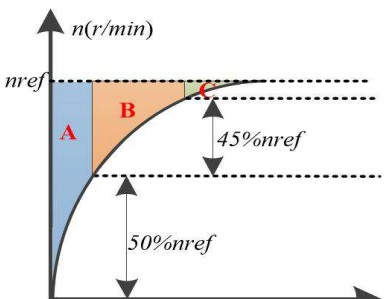

**Figure 6.** The curve of the ideal response.

In details, the dynamic response process of the controlled system can be simply divided into three stages according to the ideal response of the system (i.e., Stage A, B and C). The Stage A is called as the fast rising zone, in which the speed error signal is large and the integration process rapidly accumulates the speed error. Therefore, this stage will be the main process leading to the phenomenon of integral saturation. The Stage B is defined as the steady-state regulation zone, in which the system response transitions from dynamic regulation to steady-state process, and the integral regulator in the steady-state regulation zone still accumulates a certain speed error, which can play the role of the integral link in this stage to eliminate the system deviation. The Stage C is generally considered to be the stability zone, in which the system response is within the error band and runs stably around a given speed.

In order to realize the actual output tracking in terms of the ideal output curve, the primary tasks in Stage A have to ensure that the integral will not fall into saturation state. Hence, the logical threshold method is used to judge the speed error in real time, and the feedback compensation coefficient $\gamma$ ($\gamma < 0$) is introduced in Stage A. In Stage B, the rotation speed should be adjusted smoothly to the steady-state interval, and the traditional PI control idea can be adopted without additional compensation process. Stage C is in the steady-state interval, and the response speed can be appropriately improved at this stage. On the one hand, the effect of integral link is not obvious due to too much desaturation, and on the other hand, the response speed of the system can be significantly improved. The structure diagram of VPDPI control method proposed in this paper is shown in Figure 7.

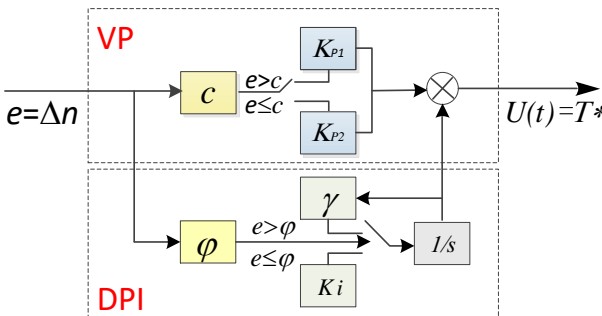

**Figure 7.** The structure diagram of VPDPI regulator control algorithm.

Where $K_{p1}$ and $K_{p2}$ are proportional coefficients, respectively. $K_i$ is the integral coefficient. In terms of the ideal response curve, *c* is the threshold used to determine whether to change the proportionality coefficient. Meanwhile, the threshold $\varphi$ is used to judge the saturation condition.

In practical application, the control parameter configuration of the proposed VPDPI regulator can usually be divided into the following steps.

(1)  completing the mathematical model equivalence of PMSM drive system;
(2)  analyzing the influence of different control parameters (such as $K_{p1}$, $K_{p2}$ and $K_i$) on the stability;
(3)  selecting the control parameters according to different drive performance requirements and observation performance;
(4)  verifying the validity of the selected parameters under different conditions.

**Remark 1.** *The specific steps for selecting threshold c, threshold $\varphi$ and the feedback compensation coefficient $\gamma$ are as follows.*

(1)  *According to the ideal curve, the threshold $c_{ref}$ should be selected around $0.95n_{ref}$. Meanwhile, the threshold $\varphi_{ref}$ should be selected near $0.5n_{ref}$.*
(2)  *The effects of different threshold values and the feedback compensation coefficient on response performance and observation performance of the system should be analyzed.*
(3)  *Fine-tune the selected threshold near the reference value $c_{ref}$ and $\varphi_{ref}$, respectively.*
(4)  *Adjust the feedback compensation coefficient $\gamma$ ($\gamma < 0$) to ensure that the cumulative error is eliminated.*
(5)  *Verify the effect of the selected thresholds under different conditions.*

It is obviously that the VPDPI regulator can be divided into VP regulator and DPI regulator. When the speed error *e* is higher than the threshold $\varphi$, the compensation coefficient $\gamma$ ($\gamma < 0$). The rest is governed by the normal integral coefficients $K_i$. Similarly, when the speed error *e* is higher than the threshold *c*, VP link moves. Meanwhile, $K_{p2}$ acts as the proportional coefficient of PI controller ($K_{p2} > K_{p1}$).

Therefore, the VPDPI control law can be expressed as shown in (10):

$$u(t) = [\rho(K_{p2} - K_{p1}) + K_{p1}]e + \gamma K_i \int edt \tag{10}$$

In (10), $\rho$ is defined the selection coefficient, and if $e > c$ then $\rho = 0$, otherwise $\rho = 1$.

### 3.2. The Stability Analysis of Regulator

Based on the transfer function block diagram in Figure 1 and the system identification results $G_{obj}$ in (7), it can be seen that the open-loop transfer function $G_{open}(s)$ of the sensorless system can be equivalent to below:

$$G_{open}(s) = \frac{K_p(s + K_i)}{s} G_{obj}(s) \tag{11}$$

According to (11), the driving system designed in this paper is equivalent to a sixth-order system, which has six open-loop poles and two open-loop zeros. Therefore, it is a better choice to use root locus analysis system stability. When the PI parameters are standardized, the open-loop transfer function of the system can be further written as follow:

$$G_{open}(s) = \frac{K_c(8.023 \times 10^{10}s^2 + 9.208 \times 10^{11}s + 8.406 \times 10^{11})}{s^6 + 1.4 \times 10^4 s^5 + 2.9 \times 10^6 s^4 + 2 \times 10^9 s^3 + 3.3 \times 10^{10} s^2 + 4.8 \times 10^7 s} \quad (12)$$

In (12), $K_c$ is the amplification factor.

The root locus of the closed system is obtained by using the root trajectory drawing rule, as shown in Figure 8. Among them, Figure 8a is the overall root trajectory, and Figure 8b,c are magnified zero-pole images.

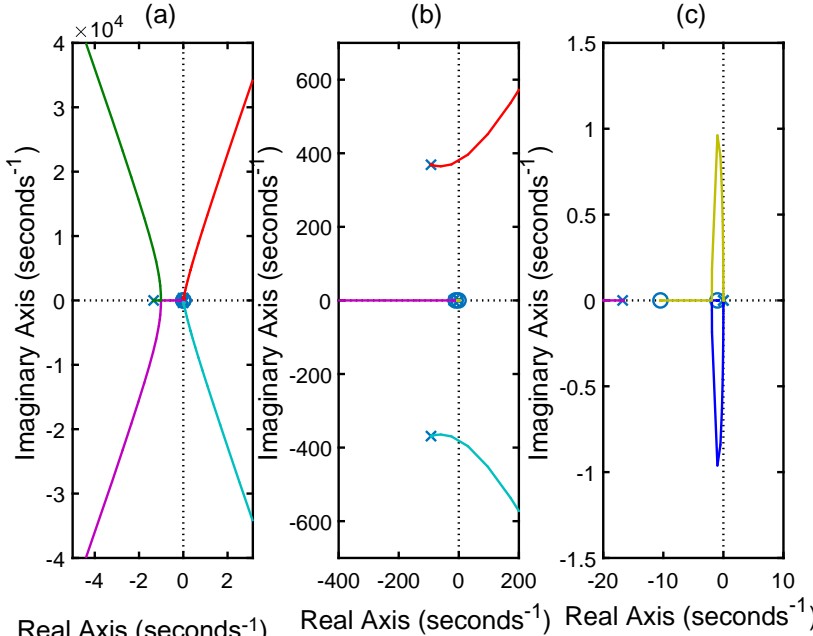

**Figure 8.** (**a**) The overall trend of the root locus. (**b**,**c**) show enlarged images of different proportions of poles, respectivelyof the sensorless system.

It is well known that, to ensure the stability of the system, all root trajectories should be restricted to the left half plane of the complex frequency domain. However, it can be seen in Figure 8b that when $K_c$ is greater than 0.45, the root locus crosses the imaginary axis and enters the right half-plane of the complex frequency domain. This means that the system is stable when the limit $K_c$ is less than the critical value.

When the integral coefficient $K_i = 1$, the Bode diagram under different proportionality coefficients is shown in Figure 9, and the corresponding amplitude-frequency margin and phase frequency margin information are shown in Table 1.

**Table 1.** The characteristics with different proportionality coefficients.

| Proportionality Coefficient $K_p$ | Magnitude Margin $G_m$ | Phase Margin $P_m$ | Stability (Yes/No) |
|---|---|---|---|
| 0 | 20.3 (at 65.8 rad/s) | 11.1 (at 18.8 rad/s) | Yes |
| 0.1 | 12.9 (at 377 rad/s) | 81 (at 41.1 rad/s) | Yes |
| 0.2 | 6.89 (at 378 rad/s) | 83.8 (at 85.5 rad/s) | Yes |
| 0.3 | 3.39 (at 378 rad/s) | 78.8 (at 140 rad/s) | Yes |
| 0.4 | 0.897 (at 379 rad/s) | 18.4 (at 349 rad/s) | Yes |
| 0.5 | −1.04 (at 379 rad/s) | −11.9 (at 398 rad/s) | No |

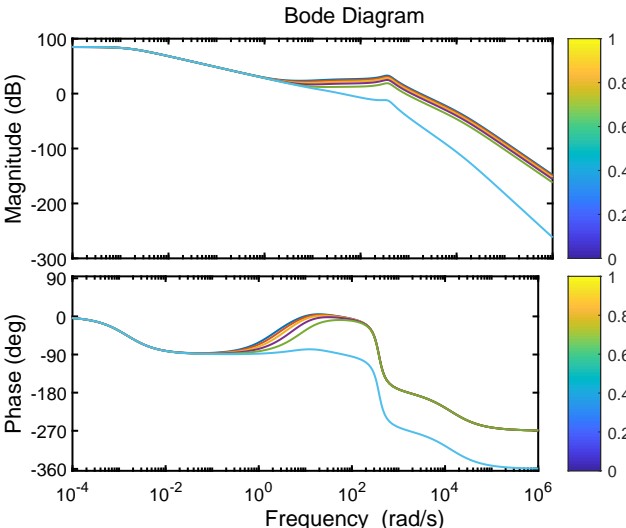

**Figure 9.** The bode diagram under the different $k_p$.

As above aforementioned results, $K_p$ has a positive phase margin in the range of $0 \sim 0.4$, and the control system is more stable. To ensure the stability of the sensorless system, the proportionality coefficient $K_{p1}$ and $K_{p2}$ should be less than 0.45 when the integration coefficient $K_i$ is set to 1. Therefore, When the median of this range $K_p = 0.2$ is selected, the different $K_i$ is selected to discuss the optimal value of $K_i$ in the selected data. The Bode diagram under different integral coefficients is shown in Figure 10, and the corresponding amplitude-frequency margin and phase frequency margin information are shown in Table 2.

In terms of the above results, when the system is purely proportional control, it is an unstable system. However, with the increase of the integral coefficient from 0 to 1, the stability margin of the system decreases slightly, but it is always stable. Therefore, the influence of proportion coefficient and integral coefficient on position sensorless system is comprehensively considered in this paper, and the final control parameters are shown in Table 3.

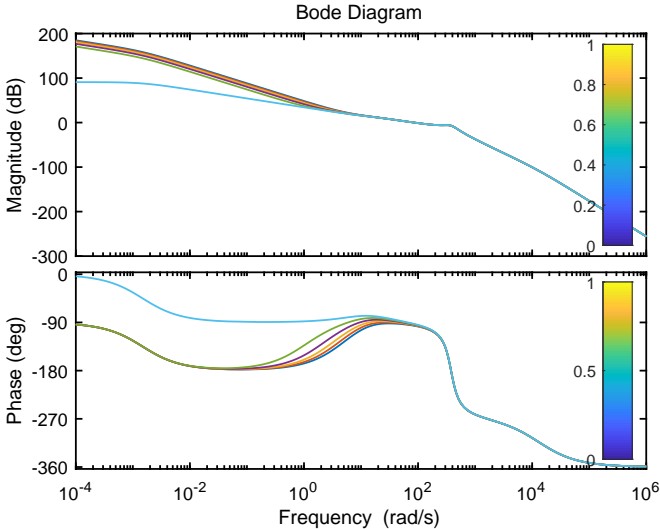

**Figure 10.** The bode diagram under the different $K_i$.

**Table 2.** The characteristics with different integral coefficients.

| Proportionality Coefficient $K_i$ | Magnitude Margin $G_m$ | Phase Margin $P_m$ | Stability (Yes/No) |
|---|---|---|---|
| 0 | 6.94(at 379 rad/s) | 87.1(at 85.7 rad/s) | No |
| 0.2 | 6.93(at 379 rad/s) | 86.5(at 85.7 rad/s) | Yes |
| 0.4 | 6.92(at 379 rad/s) | 85.8(at 85.7 rad/s) | Yes |
| 0.6 | 6.91(at 378 rad/s) | 85.1(at 85.7 rad/s) | Yes |
| 0.8 | 6.9(at 378 rad/s) | 84.5(at 85.8 rad/s) | Yes |
| 1 | 6.89(at 378 rad/s) | 83.8(at 85.5 rad/s) | Yes |

**Table 3.** The setting of regulators parameters.

| Regulator Type | Parameter | Value |
|---|---|---|
| PI | Proportionality coefficient $K_p$ | 0.2 |
|  | Integral coefficient $K_i$ | 1 |
| VPDPI | Basic proportionality coefficient $K_{p1}$ | 0.2 |
|  | Changing proportionality coefficient $K_{p2}$ | 0.4 |
|  | Integral coefficient $K_{i1}$ | 1 |
|  | Proportional switching threshold $c$ | 50 |
|  | Compensate switching threshold $\varphi$ | 500 |
|  | Compensation coefficient $\gamma$ | −14 |

## 4. Simulation Verification and Result Analysis

To demonstrate the correctness of the proposed method in the sensorless control system of PMSM, the offline simulation and the real-time experiment are used to verify the performance of the proposed method under various operating conditions, respectively.

### 4.1. Simulation Experiment Platform and Key Parameters Setting

Based on the overall structure of PMSM sensorless drive system in Section 2, the structure diagram is built as shown in Figure 11.

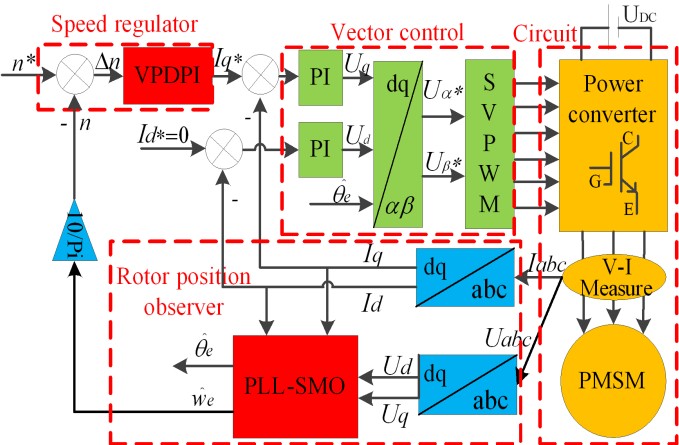

**Figure 11.** The structure diagram of sensorless PMSM drive system.

Moreover, the simulation model of sensorless drive system is built on the MATLAB/Simulink simulation platform. Meanwhile, the VPDPI regulator and the conventional PI regulator are tested under the same circuit parameters configuration, respectively. Table 3 present the parameters configuration of controller. The parameters configuration of PMSM are listed in Table 4.

**Table 4.** The setting of PMSM parameters.

| Parameter | Value |
| --- | --- |
| Stator phase resistance $R_s$ ($\Omega$) | 0.011 |
| d-axis and q-axis phase inductances $L_d$, $L_q$ (mH) | 1.6, 1 |
| Moment of inertia $J$ (kg· m$^2$) | 0.0008 |
| Rotor pole pairs $p_n$ | 3 |
| Reference speed $n_{ref}$ (rpm = r/min) | 1000 |
| Flux linkage $\psi_f$ (Wb) | 0.077 |

### 4.2. Performance Testing under Multiple Operating Conditions

It is worth noting that the sensorless drive system desires to be operated at rated conditions. However, the control system may run in an unsatisfactory mode when the influence of disturbance or fault factors are considered in the actual site. Therefore, on the basis of analyzing the performance of the motor at rated speed, it is necessary to test and analyze the performance of PMSM at low speed (80% of rated speed) and high speed (120% of rated speed). Meanwhile, the detection accuracy and stability of feedback signals have gradually become the focus to evaluate the performance of sensorless system. Therefore, the dynamic performance, steady-state performance and detection accuracy between the VPDPI regulator and conventional PI regulator are comprehensively compared and analyzed under multiple working conditions based on the SMO-PLL sensorless control structure.

#### 4.2.1. Performance Verification at Rated Speed

As shown in Table 4, all the comparative experiments are conducted on the same control parameters, and the performance can be fairly compared with different control algorithms. Meanwhile, all the simulation results are obtained under the same running operations, which refers to the load that changes from no-load to 2 N·m at 0.4 s.

When the speed is set at rated speed (1000 r/min), the speed response and the system estimated speed error using the proposed VPDPI regulator are shown in Figure 12a,b, respectively. Figure 13a,b show the speed response and the estimated speed error using the traditional PI regulator, respectively. Figures 14a and 15a record PMSM rotor position Angle information under VPDPI regulator and PI regulator, respectively. The Angle tracking error of the sensorless system under the above two regulators are shown in Figures 14b and 15b, respectively. Moreover, the output torque and three-phase current state of PMSM are shown in Figures 16 and 17, respectively.

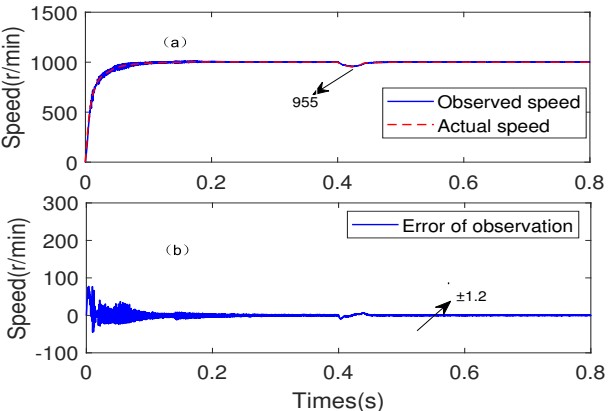

**Figure 12.** The speed response of VPDPI regulator under 1000 r/min. (**a**) Observed speed and actual speed. (**b**) Estimation error curve.

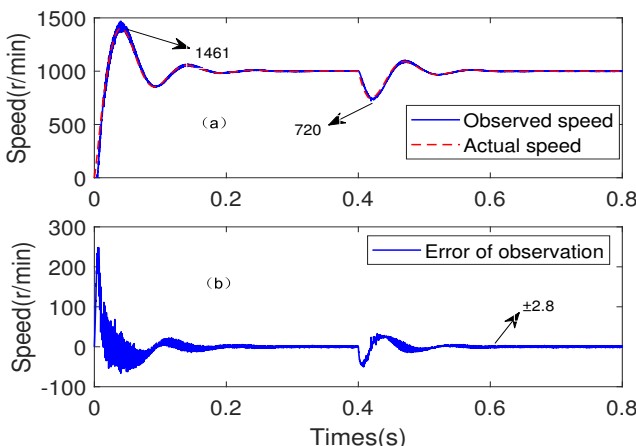

**Figure 13.** The speed response of PI regulator under 1000 r/min. (**a**) Observed speed and actual speed. (**b**) Estimation error curve.

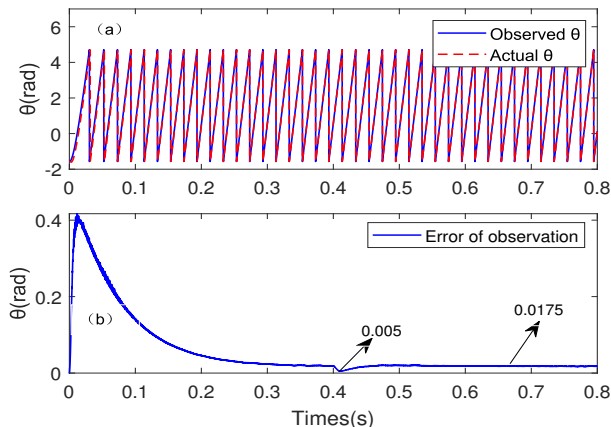

**Figure 14.** The estimation of rotor position Angle using VPDPI. (**a**) Observed and actual rotor position Angle. (**b**) Rotor position Angle observation error.

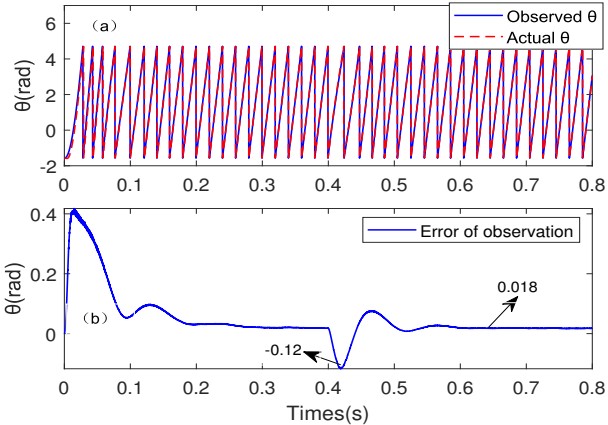

**Figure 15.** The estimation of rotor position Angle using PI. (**a**) Observed and actual rotor position Angle. (**b**) Rotor position Angle observation error.

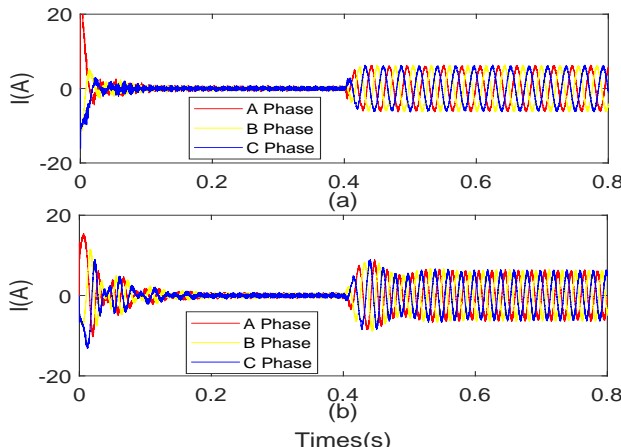

**Figure 16.** Three phase current response process under two control methods. (**a**) Current response under the proposed VPDPI control. (**b**) Current response under the PI control method

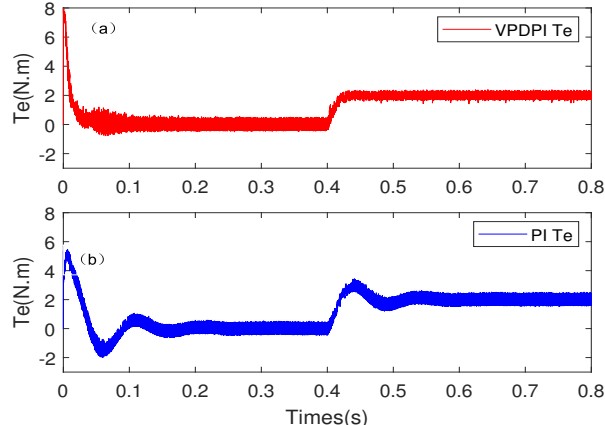

**Figure 17.** Torque response process under two control methods. (**a**) Torque response under the proposed VPDPI control. (**b**) Torque response under the PI control method

From Figures 12 and 13, It is worth noting that the VPDPI regulator has a only 0.05 s adjusting time, whereas the adjustment time of the traditional PI regulator is about 0.113 s. Meanwhile, the sensorless system is no overshoot or steady state error under the control of the VPDPI regulator. However, the overshoot can reach 461 r/min using the traditional PI regulator. On the other hand, the traditional PI regulator experiences a 280 r/min dynamic landing when the load changed, and a 45 r/min dynamic landing is happening in the VPDPI regulator. The adjustment process of sensorless system can also be further seen from the current and torque response results in Figures 16 and 17, respectively.

In the observation accuracy, the observation speed controlled by the VPDPI is about ±1.2 r/min. On the contrary, the observation speed of the traditional PI regulator has a more higher fluctuation, which even can reach ±2.8 r/min. Meanwhile, the observation error curves in Figures 14 and 15 show that the Angle tracking error on the basic of VPDPI regulator is more stable, which ensures the observation accuracy of PMSM sensorless system. In summary, the performance advantages of VPDPI regulator can be effectively verified at rated speed.

There is no doubt that dynamic performance and steady-state performance are important performance indicators of the regulating ability of the reaction regulator. However, in order to meet the complexity and uncontrollability of the actual field conditions, robustness has gradually become an important reference in measuring the performance of the regulator. Therefore, Appendix A tests the robustness of the proposed regulator at rated speed when parameters $R_s$ or $J$ do not match. Since $R_s$ is configured as 0.011 in Table 4,

VPDPI regulation performance is tested when $R_s$ is 0.0011, 0.021 and 0.031, respectively. On the other hand, since $J$ is configured as 0.0008 in Table 4, VPDPI regulation is tested when $J$ is 0.0006, 0.001 and 0.0012, respectively.

4.2.2. Performance Verification at Various Speed

The running state of the rated speed is the pursuit of the most driver systems. Due to the disturbance, it will need the sensorless system to run smoothly around the rated speed. Therefore, testing near the rated speed will be more comprehensive to verify the performance of the proposed regulator.

(1) Performance verification at 80% rated speed:

When the speed is set at 800 r/min, the estimated speed and speed error curves under two regulator are shown in Figures 18 and 19, respectively. Figures 20a,b record the response of position Angle and the response error under VPDPI regulator, respectively. The response of position Angle and the response error under VPDPI regulator are presented in Figures 21a,b, respectively. Moreover, the output torque and three-phase current state of PMSM are shown in Figures 22 and 23, respectively.

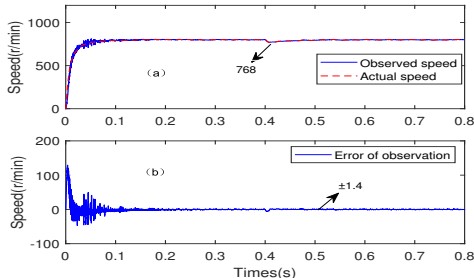

**Figure 18.** The speed response of VPDPI regulator under 800 r/min. (**a**) Observed speed and actual speed. (**b**) Estimation error curve.

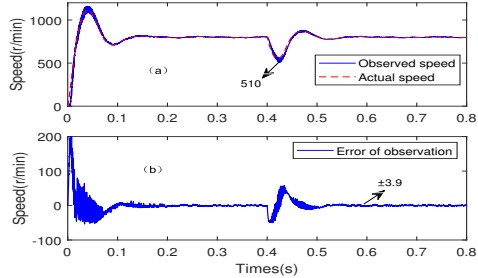

**Figure 19.** The speed response of PI regulator under 800 r/min. (**a**) Observed speed and actual speed. (**b**) Estimation error curve.

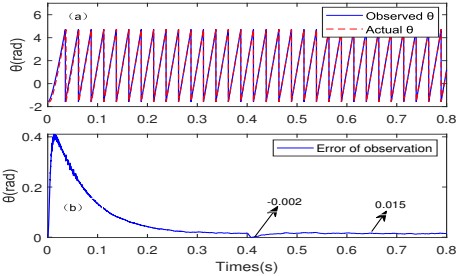

**Figure 20.** The estimation of rotor position Angle using VPDPI. (**a**) Observed and actual rotor position Angle. (**b**) Rotor position Angle observation error.

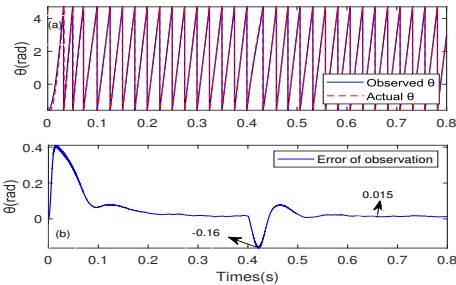

**Figure 21.** The estimation of rotor position Angle using PI. (**a**) Observed and actual rotor position Angle. (**b**) Rotor position Angle observation error.

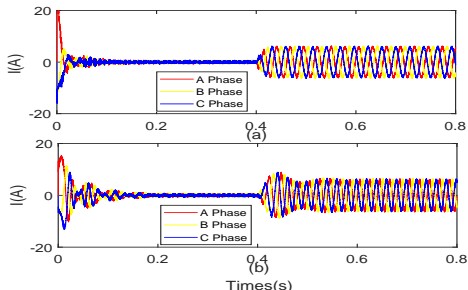

**Figure 22.** Phase current response process under two control methods. (**a**) Current response under the proposed VPDPI control. (**b**) Current response under the PI control method.

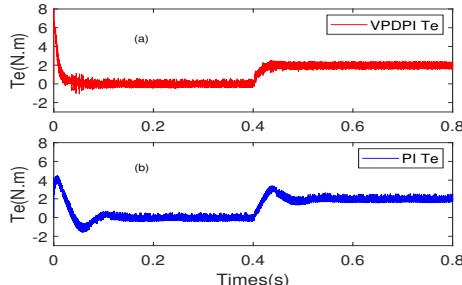

**Figure 23.** Torque response process under two control methods. (**a**) Torque response under the proposed VPDPI control. (**b**) Torque response under the PI control method.

The operation results in Figures 18 and 19 can reveal the control performance of the proposed control method from two perspectives. From the perspective of sensorless observation: the estimation errors generally exists in the starting stage. After a period of adjustment, the speed estimation errors tend to be stable. When the liability torque is added to the original system, the speed estimation error of VPDPI regulator changes little, and it soon achieves stable again. However, the control system using PI regulator once again experience an oscillation process, and it finally reaches a steady state process. From Figures 18 and 19, the proposed method controls the error range within $\pm 1.4$ r/min, and it improves the error observation accuracy by 64.1% than the traditional PI regulator ($\pm 3.9$ r/min). Therefore, the dynamic performance and steady-state performance of the traditional PI regulator have improved in the observation process, especially the suppression of overshoot.

From the perspective of speed response curve performance: the traditional PI control system has a large overshoot since the integral saturation influence. Through a adjusting process, the traditional PI control system is gradually stabilize within the error band at 0.13 s ($\Delta = 5\%$). On the contrary, the proposed VPDPI method takes into account the influence of integral saturation. The VPDPI control system is more faster and more stable entering the error band, which greatly weakens the influence of overshoot on system balance. After the load is added to the sensorless system, the dynamic landing under the

traditional PI regulator reaches 290 r/min, which go through a long period of disturbance rejection process. Unexpectedly, the control system under the VPDPI regulator only has a dynamic landing of 32 r/min, and the response quickly recovered to a stable state. Therefore, it is worth noting that the proposed method has more advantages in response curve performance analysis.

In addition to the emphasis of the observation effect, the accuracy of the observation is also an important indicator of the control of the sensorless system. As shown in Figures 20 and 21, the system estimation under the control of steady state time is similar. Nonetheless the position Angle error of the VPDPI regulator is more smoother. Meanwhile, the dynamic landing of VPDPI system is more smaller when the load is added.

In Figures 22 and 23, it can be seen that the VPDPI regulator converges to the steady-state value with a more shorter adjustment time than the traditional PI regulator.

In conclusion, when the speed is set at 800 r/min, the sensorless system using VPDPI regulator has a better dynamic performance, steady-state performance and observation effect than the traditional PI system.

(2) Performance verification at 120% rated speed:

When the speed is set at 1200 r/min, the speed and speed error curves under the control of VPDPI regulator and PI regulator are shown in Figures 24 and 25, respectively. Figures 26 and 27 record PMSM rotor position Angle information under the control of the two regulators and the error of tracking the rotor position Angle of the actual system, respectively. Moreover, the output torque and three-phase current of PMSM are shown in Figures 28 and 29, respectively.

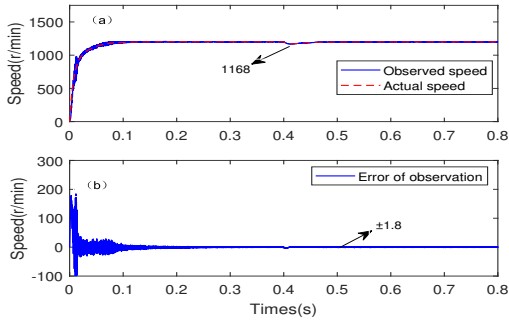

**Figure 24.** The speed response of VPDPI regulator under 1200 r/min. (**a**) Observed speed and actual speed. (**b**) Estimation error curve.

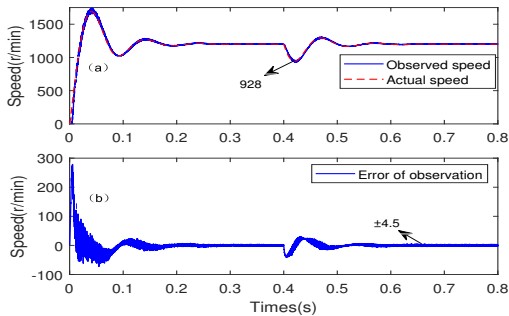

**Figure 25.** The speed response of PI regulator under 1200 r/min. (**a**) Observed speed and actual speed. (**b**) Estimation error curve.

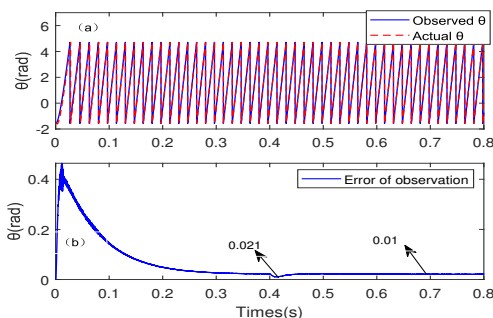

**Figure 26.** The estimation of rotor position Angle using VPDPI. (**a**) Observed and actual rotor position Angle. (**b**) Rotor position Angle observation error.

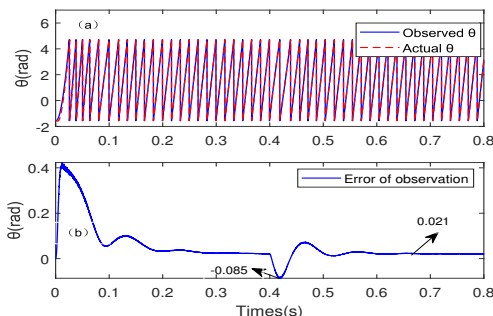

**Figure 27.** The estimation of rotor position Angle using PI. (**a**) Observed and actual rotor position Angle. (**b**) Rotor position Angle observation error.

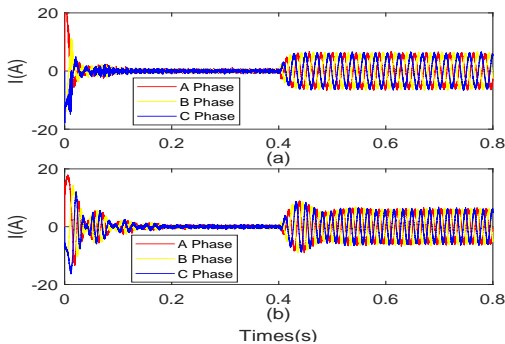

**Figure 28.** Three phase current response process under two control methods. (**a**) Current response under the proposed VPDPI control. (**b**) Current response under the PI control.

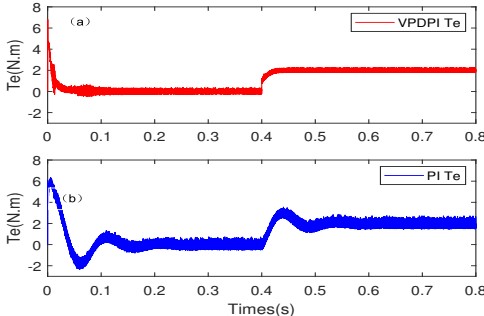

**Figure 29.** Torque response process under two control methods. (**a**) Torque response under the proposed VPDPI control. (**b**) Torque response under the PI control.

From the above experimental results, the control performance of the two regulators is similar to the above two working conditions.

The maximum overshoot of the traditional PI regulator is 548 r/min, and the adjustment time is about 0.115 s. When the load mutation occurred, a dynamic landing of 272 r/min occurred. In terms of observation accuracy, the traditional PI observation error pulsation reaches 4.5 r/min. On the contrary, the proposed VPDPI regulator does not have overshoot, and the adjustment time is only 0.053 s. The dynamic landing under load mutation is only 32 r/min. Meanwhile, the steady-state observation error fluctuation is only 1.8 r/min. The position Angle observation error is more gentle, and the observation result is more stable. Therefore, the capability of VPDPI regulator to solve integral saturation in PMSM sensorless control system is proved again.

## 5. Experimental Verification and Results

After the effectiveness of the proposed control method is tested through simulation experiments, the VPDPI control method is further loaded into the hardware-in-the-loop experimental platform to better verify the reliability of the proposed algorithm. The structure of the experimental platform is shown in Figure 30. In addition, the experimental platform is clearly presented in Figure 31. Hence, the performance differences of various control algorithms can be accurately compared. Without losing generality, the simulation control parameters are completely consistent with the experimental control parameters.

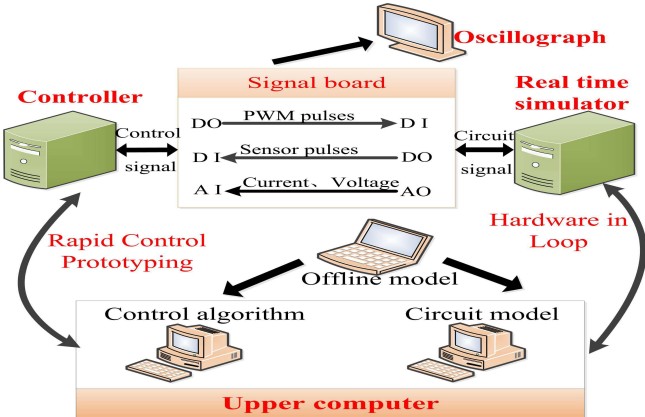

**Figure 30.** The structure of the experimental platform.

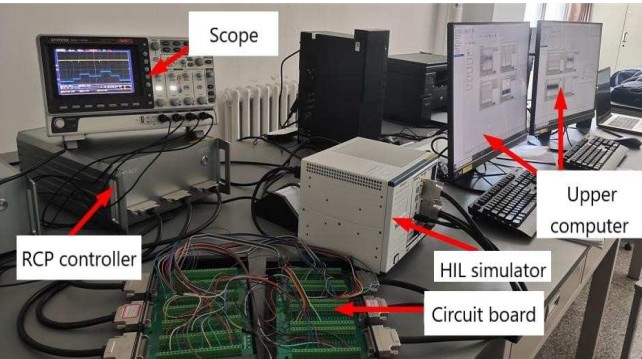

**Figure 31.** Experimental test platform for the proposed algorithm

### 5.1. Experimental Verification at Rated Speed

When the speed is set to 1000 r/min, the observed speed response of the two regulators under no-load and 2$N\dot{m}$ load are presented in Figures 32 and 33.

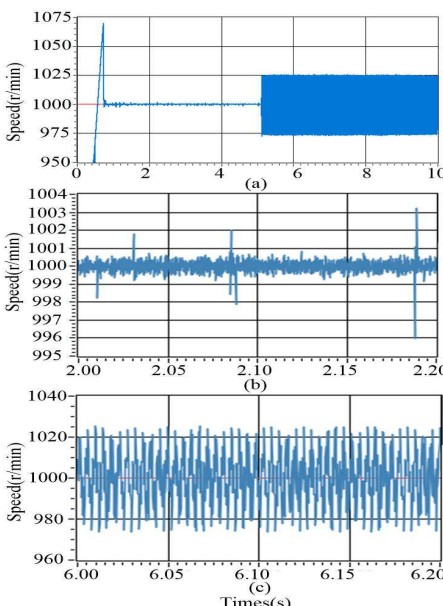

**Figure 32.** The observed speed using PI at rated speed. (**a**) The overall response diagram of observed speed at 1000 r/min. (**b**) Observed speed under no load. (**c**) Observed speed under 2 N·m load.

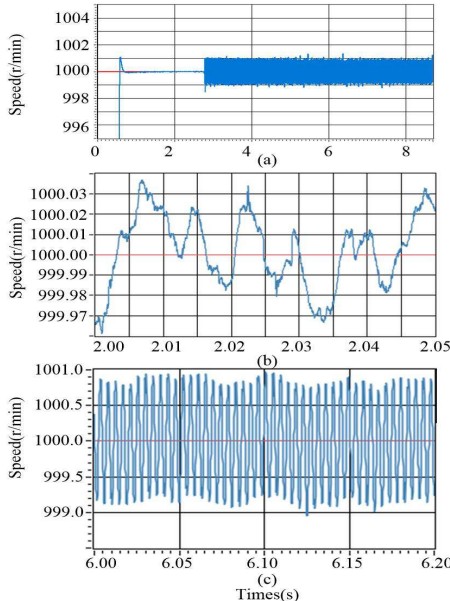

**Figure 33.** The observed speed using VPDPI at rated speed. (**a**) The overall response diagram of observed speed at 1000 r/min. (**b**) Observed speed under no load. (**c**) Observed speed under 2 N·m load.

From Figures 32 and 33, it is worth noting that the adjustment time of the proposed VPDPI regulator is clearly shorter, while almost no overshoot. However, the observed speed overshoot under the traditional PI regulator is about 75 r/min. Meanwhile, the speed pulsation of the proposed regulator is more smaller under both no-load and load conditions, especially with load condition. In addition, the observation accuracy is more higher than the traditional PI regulator when the system runs stably. Hence, the proposed regulator has outstanding dynamic performance and precision of speed estimation, which proves the validity of the method again.

### 5.2. Experimental Verification at Various Speed

(1) Experimental verification at 80% rated speed

When the speed is set to 800r/min, Figures 34 and 35 show the observed speed response of the two regulators under no-load and 2 N·m load, respectively.

From Figures 34 and 35, the observed speed under the traditional PI regulator has a large overshoot in the dynamic response process, and the result also exists a large speed ripple when the response is stable. In contrast, the maximum overshoot of the VPDPI regulator is about 0.3% of the maximum overshoot of the traditional PI. Compared with the traditional PI regulator, the speed pulsation of the proposed method is reduced about 90% under no-load condition. Moreover, the traditional PI speed pulsation becomes more serious under changing load condition, and the pulsation is up to ±25 r/min. Fortunately, the proposed method can control the pulsation within ±5 r/min, which reflects a more higher stability.

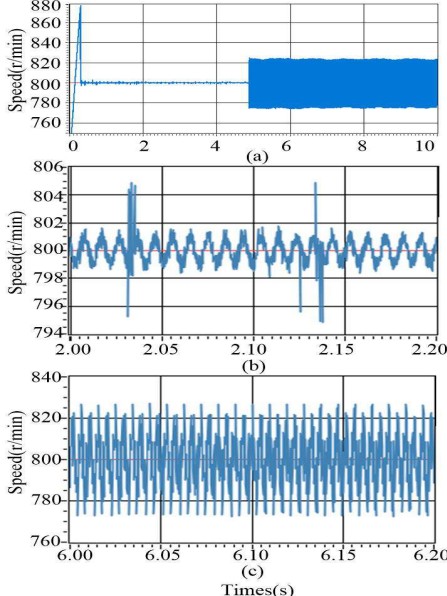

**Figure 34.** The observed speed using PI at 800 r/min. (**a**) The overall response diagram of observed speed at 800 r/min. (**b**) Observed speed under no load. (**c**) Observed speed under 2 N·m load.

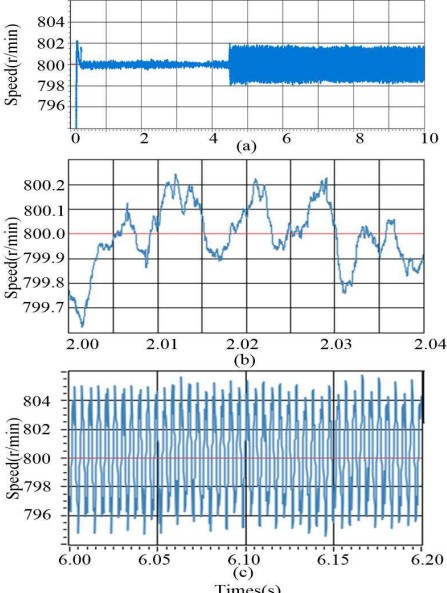

**Figure 35.** The observed speed using VPDPI at 800 r/min. (**a**) Observed speed at 800 r/min. (**b**) Observed speed under no load. (**c**) Observed speed under 2 N·m load.

(2) Experimental verification at 120% rated speed:

When the speed is set to 1200 r/min, the observed speed response of the two regulators under no-load and 2 N·m load are presented in Figures 36 and 37.

From Figures 36 and 37, it can be seen clearly that the overshoot of the traditional PI regulator is about 64 r/min. However, the overshoot of the VPDPI regulator is only 1.6% of the traditional PI regulator, which indicates a significant overshoot suppression effect. Meanwhile, the observed speed pulsations using two regulators are obviously reduced than others operation conditions, especially the VPDPI regulator. Nevertheless, it can still be detected that the pulsation of the traditional PI regulator is as high as 25 r/min with 2 N·m load condition. It is worth noting that the pulsation of VPDPI regulator is less than 0.15 r/min. In brief, the experimental results under this condition are similar to the performance reflected before.

The experimental results in this section verify the effectiveness of the proposed method to suppress integral saturation, which further improve the control performance and observation accuracy of the traditional sensorless system.

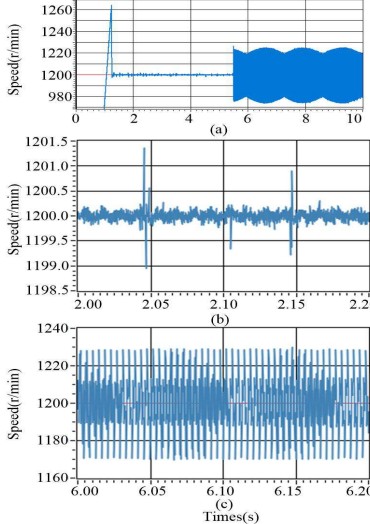

**Figure 36.** The observed speed using PI at 1200 r/min. (**a**) The overall response diagram of observed speed at 1200 r/min. (**b**) Observed speed under no load. (**c**) Observed speed under 2 N·m load.

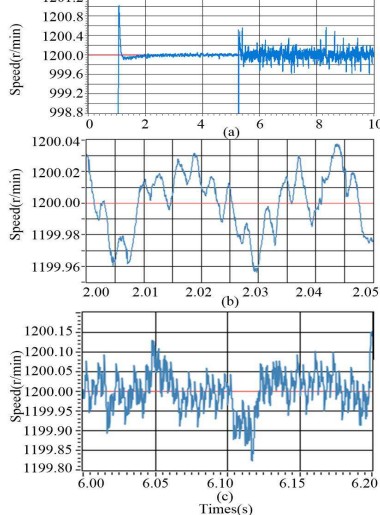

**Figure 37.** The observed speed using VPDPI at 1200 r/min. (**a**) The overall response diagram of observed speed at 1200 r/min. (**b**) Observed speed under no load. (**c**) Observed speed under 2 N·m load.

## 6. Conclusions

In this article, a VPDPI control method is creatively proposed to eliminate the integral saturation phenomenon of PMSM sensorless system, and its original contributions can be extracted as follows:

(1) The essence of integral saturation in traditional PI controller is unexpectedly revealed based on the classical control theory, and the corresponding control scheme is further developed from the perspective of dynamic response characteristic. Consequently, the novel VPDPI contains two interconnected regulators with independent effects: (a) a DPI regulator stably eliminates integral saturation by utilizing feedback compensation coefficients $\gamma$, and (b) a VP regulator ensures the system response speed by switching the proportional coefficient between $k_{p1}$ and $k_{p2}$.

(2) According to the distribution of root trajectory curve, the stability of the system is analyzed. Meanwhile, the parameters range of speed regulator is given by using frequency domain analysis method based on the identified model. Thereinto, the system has a higher stability margin when the proportional coefficient is selected in the range of $0 \sim 0.4$. When the integral coefficients are set as $0 \sim 1$, it can be found that the system is always in a stable state, but the amplitude-frequency margin and phase-frequency margin are slightly weakened.

In addition, the simulation and hardware experiments with different speed regulators are tested under the same control parameters. On the one hand, the simulation results can demonstrate the effectiveness of the new regulator under random operation conditions. Compared with the traditional PI regulator, the adjusting time of the proposed regulator is about 0.05 s under rated speed with no$-$load torque, which is less than 44.2% of the tradition PI regulator. More importantly, the overshoot is totally eliminated, which is about 461 r/min in traditional PI regulator. When the system exists a low load disturbance, the dynamic landing is reduced about 16% than that of traditional PI regulator. Moreover, the observed speed error fluctuation is also decreased about 42.9%. Similarly, the control performance remains consistent under variable speed. On the other hand, the experimental results further verify the dynamic performance, accuracy and stability of the novel regulator under multiple working conditions, especially in the suppression abilities of overshoot and the estimated speed ripple. In summary, the effectiveness and reasonability of the proposed method are both amply verified. In the future, some intelligent algorithms will be combined with the proposed method to achieve automatic adjustment of threshold parameters.

**Author Contributions:** Z.W. wrote this article, designed the control method, implemented the hardware platform, drew the figures, and performed the simulation as well as the experiment. M.Z. supervised the study, coordinated the investigations, and checked the manuscript's logical structure. X.L. validated the simulation and hardware experimental data and reviewed the writing and editing. M.L. provided practical suggestions and evaluated the feasibility of the application. All authors have read and agreed to the published version of the manuscript.

**Funding:** This work was Supported in part by the Natural Science Foundation of China under Grant No. 61563045, and in part by the International Cooperation Projects of Shihezi University under Grant GJHZ202003.

**Institutional Review Board Statement:** Not applicable.

**Informed Consent Statement:** Not applicable.

**Data Availability Statement:** Not applicable.

**Acknowledgments:** We sincerely thank Shanghai Yuankuan Energy Technology Co., Ltd. for providing technical guidance for the use of the hardware experimental platform.

**Conflicts of Interest:** The authors declare no conflict of interest.

## Abbreviations

The following abbreviations are used in this paper:

| | |
|---|---|
| *PMSM* | Permanent Magnet Synchronous Motor |
| *PI* | Proportional Integral |
| *VPDPI* | Variable-proportion Desaturation Proportional Integral |
| *ACR* | Automatic Current Regulator |
| *ASR* | Automatic Speed Regulator |
| *SNR* | Signal Noise Ratio |
| *ZPK* | Zero Pole and Gain |
| *PLL* | Phase-locked Loop |
| *SMO* | Sliding Mode Observer |
| *RCP* | Rapid Control Prototype |

## Nomenclature

The following nomenclatures are used in this manuscript:

| | |
|---|---|
| $u_d/u_q$ | stator voltage under $d/q$-axis |
| $V_d/V_q$ | observed back-EMF under $d/q$-axis |
| $i_d/i_q$ | stator current under $d/q$-axis |
| $\tilde{i}_d/\tilde{i}_q$ | observed stator current under $d/q$-axis |
| $L_d/L_q$ | stator inductance under $d/q$-axis |
| $J$ | moment of inertia |
| $R$ | stator resistance |
| $B$ | damping coefficient |
| $P_n$ | polar logarithm |
| $T_e$ | electromagnetic torque |
| $T_L$ | load torque |
| $\gamma$ | feedback compensation coefficient |
| $c$ | The judgment threshold of proportional coefficient selection |
| $\varphi$ | The judgment threshold of compensation coefficient selection |
| $\rho$ | The selection coefficient |

## Appendix A. Robustness Test Considering the Mismatch of $R_s$ or J

Figures A1–A3 show the test results when the parameters $R_s$ are not matched at rated speed. Figures A4–A6 show the test results when the coefficient $J$ is not matched.

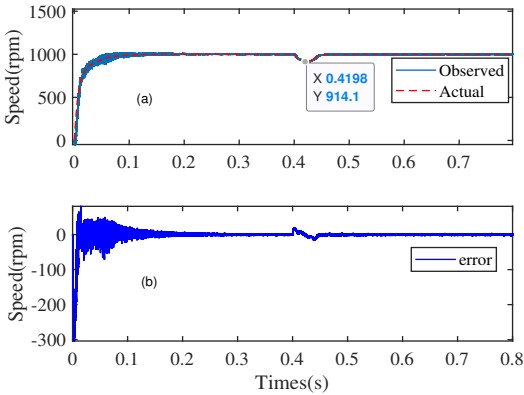

**Figure A1.** The speed response of VPDPI regulator when $R_s = 0.0011$ and $J = 0.0008$. (**a**) Observed speed and actual speed. (**b**) Estimation error curve

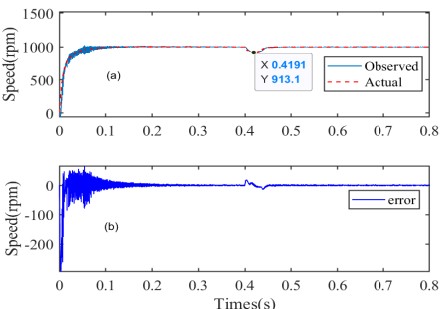

**Figure A2.** The speed response of VPDPI regulator when $R_s$ = 0.021 and $J$ = 0.0008. (**a**) Observed speed and actual speed. (**b**) Estimation error curve

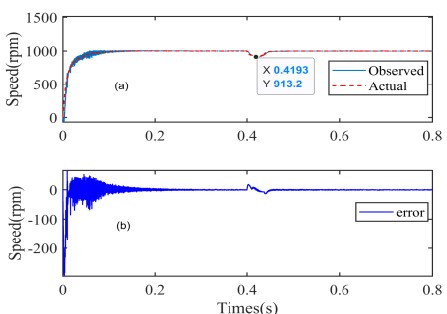

**Figure A3.** The speed response of VPDPI regulator when $R_s$ = 0.031 and $J$ = 0.0008. (**a**) Observed speed and actual speed. (**b**) Estimation error curve

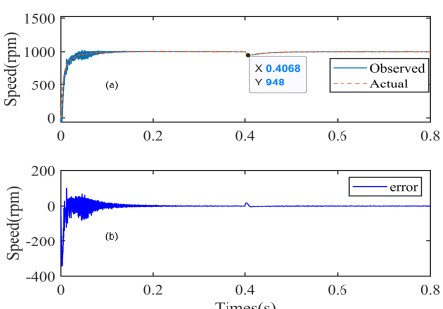

**Figure A4.** The speed response of VPDPI regulator when $J$ = 0.0006 and $R_s$ = 0.011. (**a**) Observed speed and actual speed. (**b**) Estimation error curve

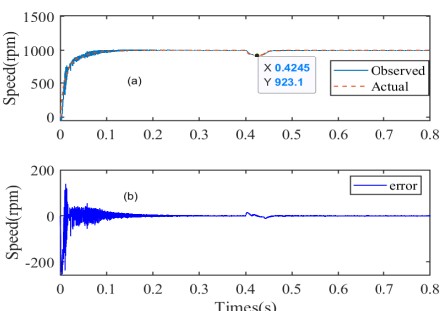

**Figure A5.** The speed response of VPDPI regulator when $J$ = 0.001 $R_s$ = 0.011. (**a**) Observed speed and actual speed. (**b**) Estimation error curve

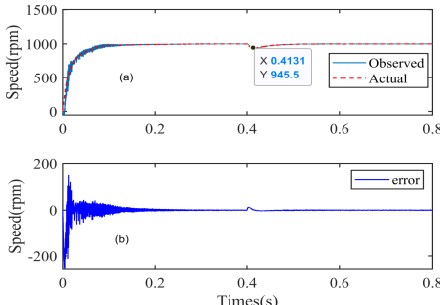

**Figure A6.** The speed response of VPDPI regulator when $J = 0.0012$ $R_s = 0.011$. (**a**) Observed speed and actual speed. (**b**) Estimation error curve

According to the test results of the appeal, it can be seen that when the parameters fluctuate, the adjustment performance of the proposed VPDPI regulator changes little, which ensures that the system without position sensor can still run stably when the parameters change, and further verifies the robustness of the proposed regulator.

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
