# Peer review of "A Novel Variable-Proportion Desaturation PI Control for Speed Regulation in Sensorless PMSM Drive System"

_applsci, doi:10.3390/app12189234_

Round 1

Reviewer 1 Report

The paper titled "A Novel Variable-proportional Desaturation PI Control for Speed Regulation in Sensorless PMSM Drive System" proposes a novel variable-proportional desaturation PI control method for PMSM drive. Comments are below.

1) The contribution of the paper must be highlighted in a better way. In particular, the major benefits of the proposed method should be highlighted in the Introduction Section.

2) Figure 1, 6 and 7 have low resolution. The quality of these figures must be improved. 

3) Other advanced control methods should be investigated to perform the comparison work. The paper below describes the model predictive control strategy for PMSM system.

"Modulated Model Predictive Control of Permanent Magnet Synchronous Motors with Improved Steady-State Performance"

The reference paper can be used to conduct a quantitative comparison work. It may improve the quality of the paper.

4) The stability analysis of the proposed method should be performed. Please mathematically prove that the closed-loop system is stable.

Author Response

Many thanks to the reviewer for your valuable opinions on our manuscript, which are very professional and specific. We have revised our manuscript item by item in accordance with the comments you provided. The detailed revision process is in the attachment.

Reviewer 2 Report

The authors presented a desaturation PI controller to be used with a sensorless PMSM drive. The proposed controller depended on adopting the desaturation process which contributes in relieving the integral saturation effect of the PI and accordingly better steady state and reduced overshot can be ensured. The authors adopts sliding mode observer-PLL for the sensorless speed estimation. Some simulation and experimentals are carried out.

My comments are as following:

- Many abbreviations are used directly without introducing their full expressions: i.e. SMO, SNR, ACR, ZPK.

- The contributions should be presented as items at the end of the introduction part.

- In line 40 and 41, the authors claim that the MRAS has strong robustness against system uncertainties and parameters variation: I think this is not accurate.

- Provide the Matlab simulink block layout used for identifying the unknown parameters of equivalent control circuit.

- Why you didn't consider the determination of Kp1, Kp2 and Ki using a mathematical derivation such as analyzing the characteristic polynomial equations of the system's transfer function: Instead of trying and checking as you did. It will be very useful to provide this analysis. I think it is easy, especially you present the bode diagrams for the change of Kp and Ki.

- As you claimed in your discussion, provide a simple robustness test considering the mismatch of Rs or J.

Author Response

(The authors gave the same response as above.)

Round 2

Reviewer 1 Report

The authors have addressed all comments. Congratulations.

Reviewer 2 Report

The authors are clearly replied to all raised criticisms. I recommends the acceptance